# Derivation and external validation of clinical prediction rules identifying children at risk of linear growth faltering

Sharia M Ahmed[1]*, Ben J Brintz[2], Patricia B Pavlinac[3], Lubaba Shahrin[4], Sayeeda Huq[4], Adam C Levine[5], Eric J Nelson[6], James A Platts-Mills[7], Karen L Kotloff[8], Daniel T Leung[1,9]*

[1]Division of Infectious Diseases, University of Utah School of Medicine, Salt lake City, United States; [2]Division of Epidemiology, University of Utah School of Medicine, Salt Lake City, United States; [3]Department of Global Health, Global Center for Integrated Health of Women, Adolescents and Children (Global WACh), University of Washington, Seattle, United States; [4]International Centre for Diarrhoeal Disease Research, Dhaka, Bangladesh; [5]Department of Emergency Medicine, Warren Alpert Medical School of Brown University, Providence, United States; [6]Department of Pediatrics and Environmental and Global Health, Emerging Pathogens Institute, University of Florida, Gainesville, United States; [7]Division of Infectious Diseases and International Health, University of Virginia, Charlottesville, United States; [8]Department of Pediatrics, Center for Vaccine Development, University of Maryland School of Medicine, Baltimore, United States; [9]Division of Microbiology & Immunology, University of Utah School of Medicine, Salt Lake City, United States

*For correspondence:
sharia.m.ahmed@utah.edu
(SMA);
daniel.leung@utah.edu (DTL)

Competing interest: The authors declare that no competing interests exist.

## Abstract

**Background:** Nearly 150 million children under-5 years of age were stunted in 2020. We aimed to develop a clinical prediction rule (CPR) to identify children likely to experience additional stunting following acute diarrhea, to enable targeted approaches to prevent this irreversible outcome.

**Methods:** We used clinical and demographic data from the Global Enteric Multicenter Study (GEMS) to build predictive models of linear growth faltering (decrease of ≥0.5 or ≥1.0 in height-for-age z-score [HAZ] at 60-day follow-up) in children ≤59 months presenting with moderate-to-severe diarrhea, and community controls, in Africa and Asia. We screened variables using random forests, and assessed predictive performance with random forest regression and logistic regression using five-fold cross-validation. We used the Etiology, Risk Factors, and Interactions of Enteric Infections and Malnutrition and the Consequences for Child Health and Development (MAL-ED) study to (1) re-derive, and (2) externally validate our GEMS-derived CPR.

**Results:** Of 7639 children in GEMS, 1744 (22.8%) experienced severe growth faltering (≥0.5 decrease in HAZ). In MAL-ED, we analyzed 5683 diarrhea episodes from 1322 children, of which 961 (16.9%) episodes experienced severe growth faltering. Top predictors of growth faltering in GEMS were: age, HAZ at enrollment, respiratory rate, temperature, and number of people living in the household. The maximum area under the curve (AUC) was 0.75 (95% confidence interval [CI]: 0.75, 0.75) with 20 predictors, while 2 predictors yielded an AUC of 0.71 (95% CI: 0.71, 0.72). Results were similar in the MAL-ED re-derivation. A 2-variable CPR derived from children 0–23 months in GEMS had an AUC = 0.63 (95% CI: 0.62, 0.65), and AUC = 0.68 (95% CI: 0.63, 0.74) when externally validated in MAL-ED.

**Conclusions:** Our findings indicate that use of prediction rules could help identify children at risk of poor outcomes after an episode of diarrheal illness. They may also be generalizable to all children, regardless of diarrhea status.

**Funding:** This work was supported by the National Institutes of Health under Ruth L. Kirschstein National Research Service Award NIH T32AI055434 and by the National Institute of Allergy and Infectious Diseases (R01AI135114).

## Editor's evaluation

This work would be valuable to global health scientists, particularly in low- and middle-income countries where childhood stunting is an ongoing challenge, and to statisticians interested in building clinical prediction rules. The authors' solid methodology leveraged large, rich datasets from multi-center studies to build and validate predictive models.

## Introduction

Despite recent advances in the prevention and treatment of childhood malnutrition, nearly 150 million children under-5 years of age were stunted in 2020 (*UNICEF et al., 2021*). Stunting is defined as a length- or height-for-age *z*-score (HAZ) 2 or more standard deviations below the population median (*de Onis et al., 2013*), and is considered both an indicator of underlying deficits (i.e., chronic malnutrition, *The World Bank, 2021*), as well as a potential contributor to future health problems (e.g., through poor immune system maturation, *Rytter et al., 2014*; *Bourke et al., 2016*). Furthermore, stunting has been consistently associated with increased risk of morbidity and mortality, delayed or deficient cognitive development, and reduced educational attainment (*McDonald et al., 2013*; *Black et al., 2013*; *Olofin et al., 2013*; *Adair et al., 2013*; *de Onis and Branca, 2016*; *Black et al., 2008*; *Bhaskaram, 2002*). Timely and accurate identification of children most likely to experience stunting offers an opportunity to prevent such negative health outcomes.

Stunting has been linked with diarrheal diseases across many settings (*Checkley et al., 2008*). An estimated 10.9% of global stunting is attributable to diarrhea (*Danaei et al., 2016*), and a child with diarrhea is more likely to have a lower HAZ or to die than age-matched controls (*Kotloff et al., 2013*). Given the 1.1 billion episodes of childhood diarrhea that occur globally every year (*Collaborators GBDDD, 2018*), assessment of children seeking healthcare for diarrhea treatment provides an opportunity to identify those at increased risk for negative outcomes, including stunting and death. Once identified, these children could be specifically targeted for intensive interventions, thereby more efficiently allocating public health resources.

In this study, we aimed to develop parsimonious, easy to implement clinical prediction rules (CPRs) to identify children under-5 most likely to experience linear growth faltering among community-dwelling children presenting to care for acute diarrhea. CPRs are algorithms that aid clinicians in interpreting clinical findings and making clinical decisions (*Reilly and Evans, 2006*). Linear growth faltering, or falling below standardized height/length growth trajectory projections, captures children whose growth has slowed precipitously and is a precursor of stunting. A number of prior studies have identified risk factors for linear growth faltering (*Danaei et al., 2016*; *Prado et al., 2019*; *Sofiatin et al., 2019*; *Naylor et al., 2015*; *Zhang et al., 2017*; *Richter et al., 2018*; *Schott et al., 2013*; *Richard et al., 2019*; *Rogawski et al., 2017*; *Rogawski et al., 2018*), but many of these were single-site studies using traditional model building approaches, some of which lacked appropriate assessments of model discrimination and calibration. Building on this body of literature, we used machine learning methods on data from two large multicenter studies to derive and externally validate prognostic prediction models for growth faltering, with the hopes of reliably identifying children that would most benefit from additional nutritional intervention after care for acute diarrhea.

## Methods

### Study population for derivation cohort 1 (GEMS)

We used data from The Global Enteric Multicenter Study (GEMS) to derive CPRs for growth faltering. The GEMS study has been described elsewhere in-depth (*Kotloff et al., 2013*; *Kotloff et al., 2012*). Briefly, GEMS was a prospective case–control study of acute moderate-to-severe diarrhea (MSD) in children 0–59 months of age. Data were collected in December 2007–March 2011 from seven sites in Africa and Asia, including those in Mali, The Gambia, Kenya, Mozambique, Bangladesh, India, and Pakistan. MSD was defined as diarrhea accompanied by one or more of the following: dysentery, dehydration, or hospital admission. Diarrhea was defined as new onset (after ≥7 days diarrhea-free) of three or more looser than normal stools in the previous 24 hr lasting 7 days or less. Cases were enrolled at initial presentation to a sentinel hospital or health center, and matched within 14 days to one to three controls without diarrhea enrolled from the community. Demographics, epidemiological, and clinical information were collected from caregivers of both cases and controls via standardized questionnaires, and clinic staff conducted physical examinations and collected stool samples which have undergone conventional and molecular testing to ascertain the pathogen that caused the diarrhea. Approximately 60 days (up to 91) after enrollment, fieldworkers visited the homes of both cases and controls to collect standardized clinical and epidemiological information and repeat anthropometry.

Children were excluded if follow-up observations occurred <49 or >91 days after enrollment, or if HAZ measurements were implausible (*Brander et al., 2019*), defined as: (1) HAZ >6 or HAZ <−6; (2) change in HAZ >3; (3) >1.5 cm loss of height from enrollment to follow-up; (4) growth of >8 or >4 cm at 49- to 60-day follow-up for children ≤6 and >6 months old, respectively; (5) growth >10 or >6 cm at 61- to 91-day follow-up for children ≤6 and >6 months old, respectively.

Parents or caregivers of participants provided informed consent, either in writing or witnessed if parents or caregivers were illiterate. The GEMS study protocol was approved by ethical review boards at each field site and the University of Maryland, Baltimore, USA. This analysis utilized publicly available data, see Data Availability statement, and as such is non-human subjects research.

### Study population for derivation and validation cohort 2 (MAL-ED)

We used the Etiology, Risk Factors, and Interactions of Enteric Infections and Malnutrition and the Consequences for Child Health and Development (MAL-ED) study to (1) re-derive the best full model, and (2) externally validate a 2-variable parsimonious version of our GEMS-derived CPR for growth faltering. MAL-ED is a longitudinal birth cohort study, and study details have been described elsewhere (*MAL-ED Network Investigators, 2014*; *Platts-Mills et al., 2014*; *Platts-Mills et al., 2015*; *Richard et al., 2014*). In brief, healthy children were enrolled within 17 days of birth and followed prospectively through 24 months of age. Children were enrolled from October 2009 to March 2012 from eight countries in Asia, Africa, and South America, including Tanzania, South Africa, Pakistan, India, Nepal, Bangladesh, Peru, and Brazil. Information on household, demographic, and clinical data from mother and child were collected at enrollment and reassessed periodically, and illness and feeding information was collected at twice-weekly household visits.

In MAL-ED, diarrhea was defined as maternal report of three or more loose stools in a 24-hr period, or one loose stool with blood. Each diarrhea episode had to be separated by at least 2 days without diarrhea in order to qualify as distinct diarrhea episodes. To match MAL-ED longitudinal cohort active surveillance data to GEMS, in which children were enrolled upon presentation to clinic with acute diarrhea, we linked anthropometric measurements and other predictor variables with diarrhea episodes in MAL-ED using the following methods (https://github.com/LeungLab/CPRgrowthfaltering): First, each episode of diarrhea was linked to the closest HAZ measurement from before the onset of diarrhea symptoms, but no more than 31 days beforehand. Each diarrhea episode was also linked with the HAZ measurement closest to 75 days after the onset of diarrhea symptoms, but within 49 and 91 days inclusive. Second, each diarrhea episode was linked to the closest observation of each potential predictor variable. Each dietary intake variable had to be observed within 90 days of the diarrhea episode, and each household descriptor variable had to be observed within 6 months of the onset of diarrhea in order to be eligible, otherwise those predictors were considered missing for that specific diarrhea episode. Finally, data were split into age categories, and only one diarrhea episode per enrolled child per model was randomly selected without replacement for analysis.

The same inclusion/exclusion criteria were applied as listed above for the GEMS growth faltering analysis, with the exception that the allowed follow-up period extended up to and including 95 days.

Parents or caregivers of participants provided informed consent. The MAL-ED study protocol was approved by ethical review boards at each field site and the Johns Hopkins Institutional Review Board, Baltimore, USA. This analysis utilized publicly available data, see Data Availability statement, and as such is non-human subjects research.

## Outcomes

We defined growth faltering as a decrease in HAZ of ≥0.5 HAZ within 49–91 days of enrollment in GEMS, or within 49–95 days in MAL-ED.

## Predictive variables

In GEMS, potential predictors included over 130 descriptors of the child, household, and community, collected at enrollment (*Supplementary file 1*). Collinear or conceptually similar predictors were removed from consideration to maximize model utility (e.g., HAZ, but not MUAC was considered in the main model). We considered individual components of household wealth, but did not explore the composite wealth variable used in other reports (*Brander et al., 2019*) since its utilization in a CPR would require collecting multiple parameters that were already being considered individually.

In MAL-ED, we considered 60 potential predictors of growth faltering (*Supplementary file 1*). We limited possible predictor variables to those that would be easily assessable upon presentation to clinic in a low-resource setting (i.e., did not consider characteristics that required diagnostic testing), and again only considered individual components of combination indicators (e.g., wealth index, Vesikari score).

## Statistical analysis

In our complete-case analysis, we screened variables using variable importance measures from random forests to identify the most predictive variables. Random forests are an ensemble learning method whereby multiple decision trees (1000 throughout this analysis) are built on bootstrapped samples of the data with only a random sample of potential predictors considered at each split, thereby decorrelating the trees and reducing variability (*James et al., 2013*). Throughout this analysis, the number of variables considered at each split was equal to the square root of the total number of potential variables, rounded down. Variables were ranked by predictive importance based on the reduction in mean squared prediction error achieved by including the variable in the predictive model on out-of-bag samples (i.e., observations not in the bootstrapped sample).

Generalizable performance was assessed using fivefold repeated cross-validation. In each of 100 iterations, random forests were fit to a training dataset (random 80% sample of analytic dataset), and variable were ranked using the random forest importance measure as above. Separate logistic regression and random forest regression models were then fit to a subset of the top predictive variables in the training dataset. Subsets examined were the top 1–10, 15, 20, 30, 40, and 50 predictors. Each of these models was then used to predict the outcome (growth faltering) on the test dataset. Model performance was assessed using the receiver operating characteristic (ROC) curves and the cross-validated *C*-statistic (area under the ROC curve, AUC). The AUC describes how well a model can discriminate between a binary outcome in the test data from the cross-validated folds.

Calibration refers to a model's ability to correctly estimate the risk of the outcome (*Steyerberg and Vergouwe, 2014*). We assessed model calibration both quantitatively and graphically ('weak' and 'moderate' calibration, respectively, *Van Calster et al., 2019*). First, we assessed calibration-in-the-large, or calibration intercept, by using logistic regression to estimate the mean while subtracting out the estimate (model the log-odds of the true status, offset by the CPR-predicted log-odds). Next, we used calibration slope to assess the spread of the estimated probabilities, whereby we fit a logistic regression model with log-odds of the true status as the dependent variable and CPR-predicted log-odds as the independent variable. Finally, we assessed moderate calibration graphically, whereby we calculated the predicted probability of growth faltering for each child in a given analysis using each iteration of each *n*-variable model fit. These predicted probabilities were then binned into deciles, and the proportion of each decile who truly experienced the outcome was calculated for each iteration of each *n*-variable model. The mean predicted probability and observed proportion were calculated for

each decile across iterations. These average observed proportions were then plotted against averaged deciles for each *n*-variable model fit (see https://github.com/LeungLab/CPRgrowthfaltering for full analytic code).

Based on top predictors available in both GEMS and MAL-ED (see Results), the 2-variable GEMS-derived CPR of growth faltering was externally validated in MAL-ED data. A logistic regression was fit to all diarrhea cases age 0–23 months in GEMS data, with predictors chosen based on random forest. This model was then used to predict growth faltering in diarrhea cases in MAL-ED (age in MAL-ED converted from days to months), and discrimination and calibration were assessed as described above.

### Sensitivity and subgroup analyses

We undertook additional sensitivity and subgroup analyses to explore if our ability to predict growth faltering improved in specific patient populations or with additional predictors within GEMS data. First, we explored age-strata specific CPRs for children 0–11, 12–23, 0–23, and 24–59 months. Second, we explored the predictive ability of MUAC instead of and in addition to HAZ. Third, we attempted to account for potential seasonal variation by adding a predictor for month of diarrhea. Fourth, we added indicator variables for the use of antibiotics before presentation (enrollment), while at clinic, prescription to take home after care, and ever. Fifth, we limited our outcome to only very severe growth faltering, defined as a decrease ≥1.0 HAZ (as opposed to ≥0.5 HAZ in the main analysis). Sixth, we explored the impact diarrhea etiology had on growth faltering prediction. We added variables for the presence/absence of *Shigella*, *Cryptosporidium*, *Shigella* + *Cryptosporidium* infections, and any viral etiology (defined as infection of any of the following: astrovirus, norovirus GII, rotavirus, sapovirus, and adenovirus 40/41). Etiology-specific infection were defined as an episode-specific attributable fraction (AFe) greater than or equal to a given cutoff (0.3, 0.5, and 0.7 were considered) (*Kotloff et al., 2013*). Seventh, we explored the prevalence of growth faltering in healthy controls, and identified top predictors and their ability to predict growth faltering in controls. Potential predictors related to diarrhea were not considered amongst controls (e.g., number of days with diarrhea at presentation). Eighth, we explored the role of stunting at presentation on growth faltering by limiting the CPR to only those who were not already stunted at presentation (HAZ ≥−2). Ninth, we fit a CPR predicting any stunting at follow-up, both among all presenting patients as well as limited to those NOT stunted at presentation. Finally, we conducted a quasi-external validation within the GEMS data by fitting a model to one continent and validating it on the other. All analysis was conducted in R 4.0.2 using the packages 'ranger', 'cvAUC', and 'pROC'.

## Results

### Growth faltering in children following acute diarrhea in GEMS and MAL-ED

There were 9439 children with acute diarrhea enrolled in GEMS. In the analysis of the primary outcome (growth faltering), 110 observations were dropped for having follow-up measurements taken <49 or >91 days after enrollment, and 1276 were dropped for having implausible HAZ measurements, leaving an analytic sample of 8053. An addition 414 observations were dropped for having missing predictor data, as random forest analysis requires complete cases. Of the remaining 7639 children, 1744 (22.8%) experienced severe growth faltering (≥0.5 decrease in HAZ), and 357 (4.7%) experienced very severe growth faltering (≥1.0 decrease in HAZ) (*Figure 1*). Growth faltering rates differed by country, with Mozambique and The Gambia having the highest rates of growth faltering (34.5% and 31.9% experienced severe growth faltering, respectively) and Mali having the lowest rate (14.9%, *Supplementary file 1*). Growth faltering rates also varied by child's age, with a higher proportion of younger children experiencing growth faltering than older children (*Supplementary file 1*).

In the analysis of MAL-ED data, we started with 6617 diarrhea episodes from 1390 children. In order to align with GEMS inclusion criteria and limit to acute onset diarrhea, 566 diarrhea episodes were dropped for having prolonged or persistent diarrhea (>7 days duration). An additional 125 episodes were dropped for having missing HAZ measurements or an HAZ follow-up measurement <49 or >95 days from diarrhea onset, and 138 episodes were dropped for having implausible HAZ measurements, leaving 5788 diarrhea episodes from 1350 children. An additional 105 observations were dropped for having missing predictor data. Of the remaining 5683 observations from 1322 children, 961 (16.9%)

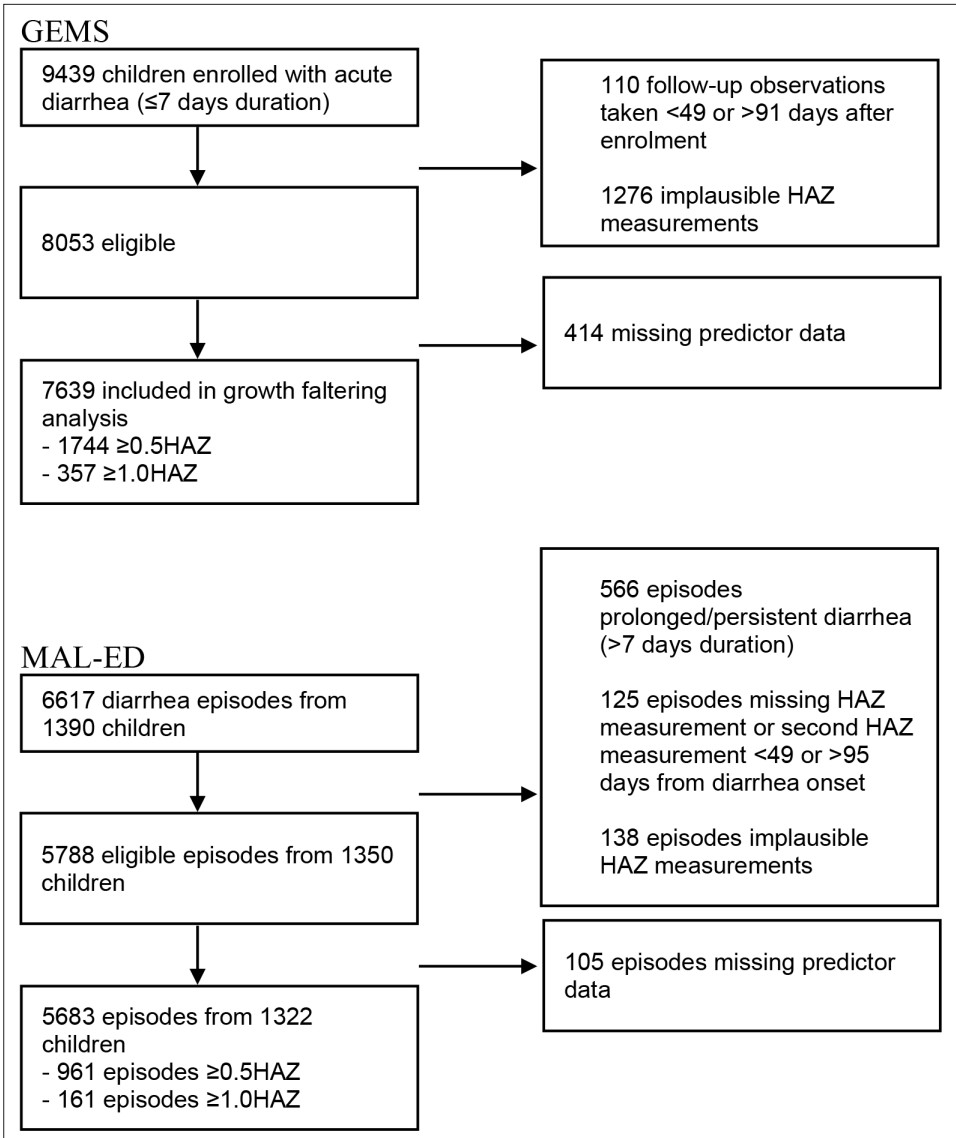

**Figure 1.** Flow diagram of study inclusion.

episodes experienced severe growth faltering (≥0.5 decrease in HAZ) and 161 (2.8%) episodes experienced very severe growth faltering (≥1.0 decrease in HAZ, *Figure 1*).

## Derivation of a CPR to identify children who went on to severe growth faltering following acute diarrhea using GEMS data

After random forest screening of variables, logistic regression models consistently had higher AUCs than random forest regression models (*Figure 2*), therefore we only present the easier to interpret logistic regression results moving forward. In *Table 1*, we show the top 10 most predictive variables ranked from most to least important, for severe growth faltering (≥0.5 decrease in HAZ). The top predictive variables for severe growth faltering were: age, HAZ at enrollment, respiratory rate, temperature, number of people living in the household, number of people sleeping in the household, number of days of diarrhea at presentation, number of other households that share same fecal waste disposal facility (e.g., latrine), whether the child was currently breastfed at time of diarrhea, and the number of children <60 months old living in the household. The maximum AUC attained with the model was 0.75 (95% confidence interval [CI]: 0.75, 0.75) with a model of 20 variables, while an AUC of 0.71 (95% CI: 0.71, 0.72), 0.72 (95% CI: 0.72, 0.72), and 0.72 (95% CI: 0.72, 0.72) could be obtained

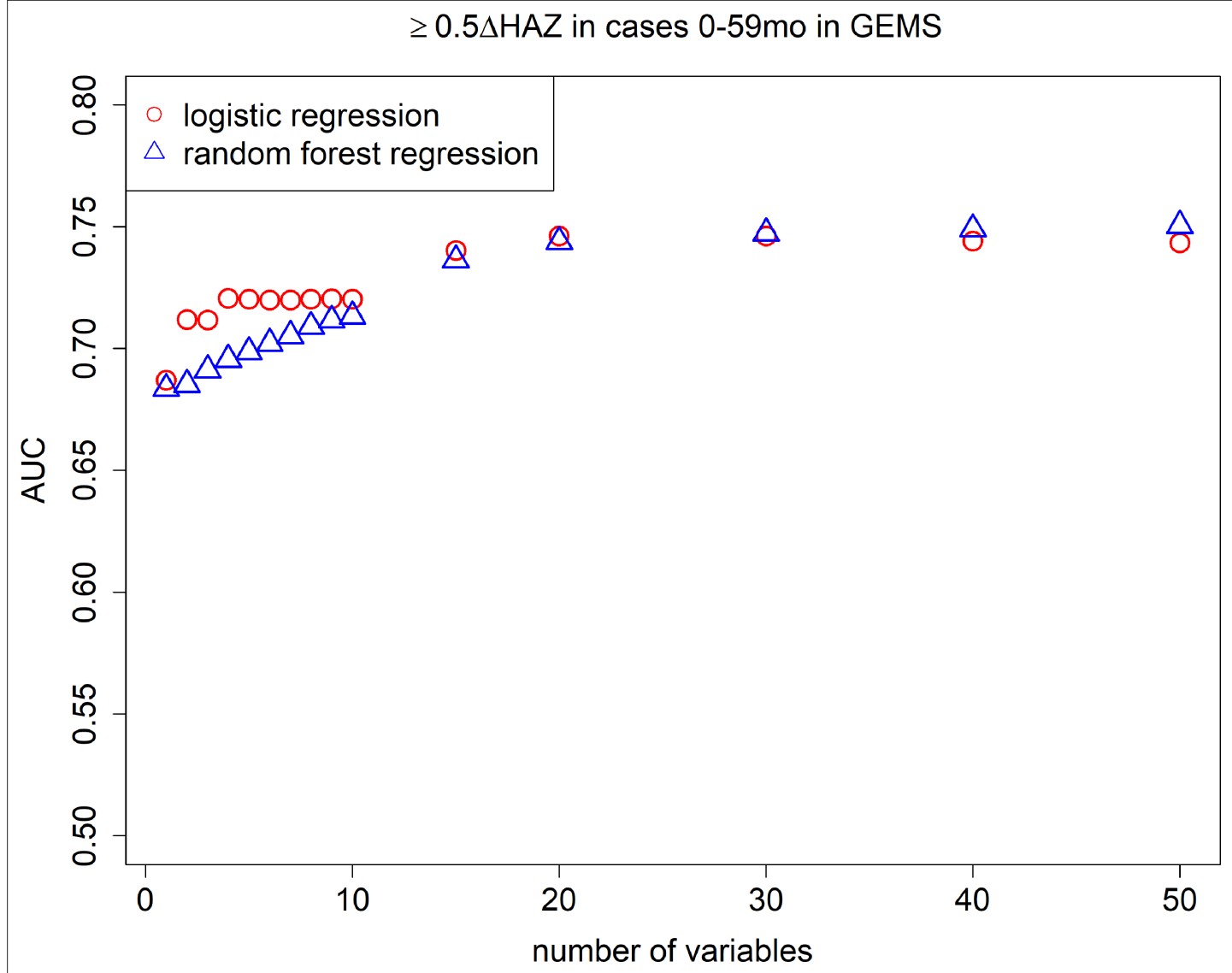

**Figure 2.** Area under the curves (AUCs). Cross-validated AUC achieved by number of predictive variables included in random forest regression and logistic regression models predicting growth faltering (≥0.5 decrease in height-for-age z-score [HAZ]) in children 0–59 months of age presenting with diarrhea in the Global Enteric Multicenter Study (GEMS).

The online version of this article includes the following figure supplement(s) for figure 2:

**Figure supplement 1.** Receiver operating characteristic (ROC) curves: average ROC curves from the cross-validated logistic regression models predicting growth faltering with 2, 5, and 10 predictors.

with a CPR of 2, 5, and 10 variables, respectively (*Figure 2*). When limited to children 0–23 months of age, AUC decreased to 0.64 (95% CI: 0.64, 0.64) for 10 variables. In the full 10-variable model, we achieved a specificity of 0.47 at a sensitivity of 0.80 (*Figure 2—figure supplement 1*). The average predicted probability of growth faltering was consistently close to the average observed probability (calibration-in-the-large, or intercept), and the spread of predicted probabilities was similar to the spread of observed probabilities (calibration slope) for models including 1–10 predictor variables (*Table 2*, *Figure 3*, *Figure 3—figure supplement 1*).

**Table 1.** Growth faltering.

Variable importance ordering and cross-validated average overall area under the curve (AUC) and AUC by patient subset and 95% confidence intervals for a 5 (bold) and 10 (italicized) variable logistic regression model for predicting growth faltering in children in 7 LMICs(Low- and middle-income countries) derived from Global Enteric Multicenter Study (GEMS) data (≥0.5 decrease in height-for-age z-score [HAZ] in children with acute diarrhea).

| Patient subset | GEMS | | | | MAL-ED | |
|---|---|---|---|---|---|---|
| | 0–59 months (main text model) | 0–59 months (limit to only those NOT stunted at beginning (HAZ >=−2) 5659/7639 (74.1%)) | 0–59 months limited to only those NOT stunted at beginning outcome is ANY stunting at follow-up (HAZ <−2) | 0–23 months (for external validation) | Healthy controls | 0–23 months |
| AUCs | **0.72 (0.72, 0.72)** | **0.71 (0.70, 0.72)** | **0.90 (0.89, 0.91)** | **0.64 (0.63, 0.65)** | **0.79 (0.78, 0.79)** | **0.67 (0.67, 0.68)** |
| | *0.72 (0.72, 0.72)* | *0.71 (0.70, 0.72)* | *0.90 (0.89, 0.90)* | *0.64 (0.64, 0.64)* | *0.79 (0.79, 0.79)* | *0.68 (0.67, 0.69)* |
| 1 | Age (months) | Age (months) | HAZ | HAZ | Age (months) | HAZ |
| 2 | HAZ | HAZ | Age | Age (months) | HAZ | Age (days) |
| 3 | Respiratory rate | Respiratory rate | Respiratory rate | Respiratory rate | Respiratory rate | Total days breastfeeding |
| 4 | Temperature | Temperature | Temperature | Temperature | Temperature | Total days in all diarrheal episodes to date |
| 5 | Num. people living in household | Num. people living in household | Num. people living in household | Num. people living in household | Num. people living in household | Mean number of people per room |
| 6 | Num. rooms used for sleeping | Num. rooms used for sleeping | Num. days of diarrhea at presentation | Num. rooms used for sleeping | Breastfed | Days with diarrhea so far in this episode |
| 7 | Num. days of diarrhea at presentation | Num. days of diarrhea at presentation | Num. other households that share same fecal waste facility | Num. days of diarrhea at presentation | Num. rooms used for sleeping | Maternal education (years) |
| 8 | Num. other households that share same fecal waste facility | Breastfed | Num. rooms used for sleeping | Num. other households that share same fecal waste facility | Num. children <60 months live in household | Days since last diarrhea episode |
| 9 | Breastfed | Num. other households that share same fecal waste facility | Num. children <60 months live in household | Num. children <60 months live in household | Caregiver education | People sleeping in house |
| 10 | Num. children <60 months live in household | Num. children <60 months live in household | Caregiver education | Caregiver education | Num. other households share latrine | Max loose stools in this episode |

**Table 2.** Calibration intercept and slope.

| Number of predictor variables | GEMS 0–59 months Intercept (95% CI) | GEMS 0–59 months Slope (95% CI) | GEMS 0–23 months (for external validation) Intercept (95% CI) | MAL-ED 0–23 months Rederivation Slope (95% CI) | MAL-ED 0–23 months Rederivation intercept (95% CI) | MAL-ED 0–23 months Rederivation Slope (95% CI) | GEMS-derived model applied to MAL-ED data Intercept (95% CI) | GEMS-derived model applied to MAL-ED data Slope (95% CI) |
|---|---|---|---|---|---|---|---|---|
| 1 | $2.9 \times 10^{-3}$ $(-1.2 \times 10^{-1}, 1.3 \times 10^{-1})$ | 1.0 (0.82, 1.2) | $-1.0 \times 10^{-2}$ (−0.14, 0.12) | 0.97 (0.62, 1.3) | $9.6 \times 10^{-3}$ (−0.32, 0.32) | 1.0 (0.35, 1.7) | | |
| 2 | $3.6 \times 10^{-3}$ $(-1.2 \times 10^{-1}, 1.3 \times 10^{-1})$ | 1.0 (0.84, 1.2) | $-1.1 \times 10^{-2}$ (−0.14, 0.12) | 0.98 (0.70, 1.3) | $1.1 \times 10^{-2}$ (−0.33, 0.33) | 1.0 (0.51, 1.5) | −0.32 (−0.54, −0.11) | 1.5 (1.0, 2.1) |
| 3 | $3.6 \times 10^{-3}$ $(-1.2 \times 10^{-1}, 1.3 \times 10^{-1})$ | 1.0 (0.84, 1.2) | $-1.2 \times 10^{-2}$ (−0.14, 0.12) | 0.97 (0.70, 1.2) | $1.1 \times 10^{-2}$ (−0.33, 0.33) | 0.99 (0.51, 1.5) | | |
| 4 | $4.1 \times 10^{-3}$ $(-1.2 \times 10^{-1}, 1.3 \times 10^{-1})$ | 1.0 (0.84, 1.2) | $-1.2 \times 10^{-2}$ (−0.14, 0.12) | 0.97 (0.71, 1.2) | $1.1 \times 10^{-2}$ (−0.33, 0.33) | 0.97 (0.49, 1.5) | | |
| 5 | $4.2 \times 10^{-3}$ $(-1.2 \times 10^{-1}, 1.3 \times 10^{-1})$ | 1.0 (0.83, 1.2) | $-1.2 \times 10^{-2}$ (−0.14, 0.12) | 0.96 (0.70, 1.2) | $1.1 \times 10^{-2}$ (−0.33, 0.33) | 0.95 (0.48, 1.5) | | |
| 6 | $4.2 \times 10^{-3}$ $(-1.2 \times 10^{-1}, 1.3 \times 10^{-1})$ | 1.0 (0.83, 1.2) | $-1.2 \times 10^{-2}$ (−0.14, 0.12) | 0.96 (0.70, 1.2) | $1.2 \times 10^{-2}$ (−0.33, 0.33) | 0.94 (0.47, 1.5) | | |
| 7 | $4.3 \times 10^{-3}$ $(-1.2 \times 10^{-1}, 1.3 \times 10^{-1})$ | 1.0 (0.83, 1.2) | $-1.2 \times 10^{-2}$ (−0.14, 0.12) | 0.96 (0.70, 1.2) | $1.2 \times 10^{-2}$ (−0.33, 0.33) | 0.92 (0.47, 1.4) | | |
| 8 | $4.4 \times 10^{-3}$ $(-1.2 \times 10^{-1}, 1.3 \times 10^{-1})$ | 1.0 (0.83, 1.2) | $-1.2 \times 10^{-2}$ (−0.14, 0.12) | 0.95 (0.69, 1.2) | $1.2 \times 10^{-2}$ (−0.33, 0.33) | 0.92 (0.47, 1.4) | | |
| 9 | $4.7 \times 10^{-3}$ $(-1.2 \times 10^{-1}, 1.3 \times 10^{-1})$ | 1.0 (0.83, 1.2) | $-1.2 \times 10^{-2}$ (−0.14, 0.12) | 0.95 (0.69, 1.2) | $1.2 \times 10^{-2}$ (−0.33, 0.33) | 0.91 (0.47, 1.4) | | |
| 10 | $4.8 \times 10^{-3}$ $(-1.2 \times 10^{-1}, 1.3 \times 10^{-1})$ | 1.0 (0.83, 1.2) | $-1.2 \times 10^{-2}$ (−0.14, 0.12) | 0.93 (0.69, 1.2) | $1.2 \times 10^{-2}$ (−0.33, 0.33) | 0.89 (0.46, 1.4) | | |

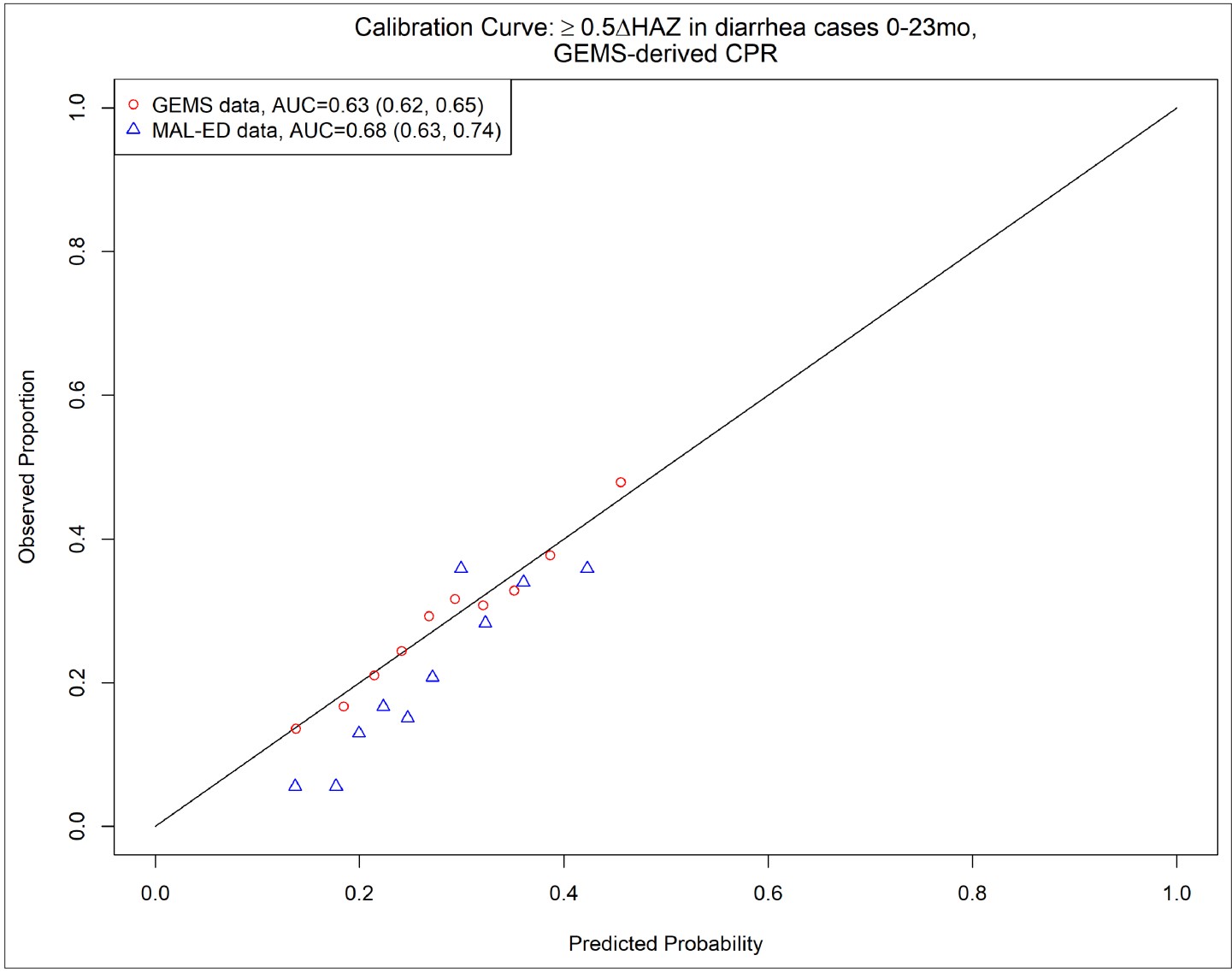

**Figure 3.** 2-Variable clinical prediction rule (CPR) for growth faltering: calibration curve and discriminative ability of 2-varaible (age, height-for-age z-score [HAZ] at enrollment) model predicting growth faltering (≥0.5 decrease in HAZ) in children presenting for acute diarrhea in LMICs.

The online version of this article includes the following figure supplement(s) for figure 3:

**Figure supplement 1.** Calibration curve of 5- and 10-variable model predicting growth faltering (≥0.5 decrease in height-for-age z-score [HAZ]) in children 0–59 months of age presenting for acute diarrhea in the Global Enteric Multicenter Study (GEMS).

**Figure supplement 2.** Area under the curves (AUCs): cross-validated AUC achieved by number of predictive variables included in random forest regression and logistic regression models predicting growth faltering (≥0.5 decrease in height-for-age z-score [HAZ]) in children 0–23 months of age presenting with diarrhea in the MAL-ED (the Etiology, Risk Factors, and Interactions of Enteric Infections and Malnutrition and the Consequences for Child Health and Development) study.

**Figure supplement 3.** Calibration curves of 5- and 10-variable model predicting growth faltering (≥0.5 decrease in height-for-age z-score [HAZ]) in children 0–23 months of age presenting for acute diarrhea in the MAL-ED (the Etiology, Risk Factors, and Interactions of Enteric Infections and Malnutrition and the Consequences for Child Health and Development) study.

**Figure supplement 4.** Area under the curves (AUCs): cross-validated AUC achieved by number of predictive variables included in random forest regression and logistic regression models predicting growth faltering (≥0.5 decrease in height-for-age z-score [HAZ]) in children 0–59 months of age without diarrhea in the Global Enteric Multicenter Study (GEMS).

**Figure supplement 5.** Histogram comparing baseline height-for-age z-score (HAZ) between children who did and did not experience growth faltering (≥0.5 decrease in HAZ) in Global Enteric Multicenter Study (GEMS) data.

## Rederivation and external validation of a CPR to identify children who went on to severe growth faltering following acute diarrhea using MAL-ED data

We then derived a CPR for growth faltering using MAL-ED data, and found that the top predictors were similar to those identified using GEMS data, with age and HAZ at diarrhea being the top 2 predictors. Other top predictors in MAL-ED included breastfeeding, total days in all diarrhea episodes, mean number of people per room of home, days with diarrhea so far in this episode, number of years of maternal education, days since last diarrhea episode, number of people sleeping in house, and loose stools in this diarrhea episode (*Table 1*). The discriminative performance of the full model was similar to that found with GEMS (0.72 [95% CI: 0.72, 0.72] in GEMS, 0.68 [95% CI: 0.67, 0.69] in MAL-ED). The average predicted probability of growth faltering was consistently close to the average observed probability (calibration-in-the-large, or intercept). The spread of predicted probabilities (calibration slope) was slightly more extreme than observed probabilities, but there was no evidence they were different than 1.0 for models including 1–10 predictor variables (slope point estimates all 95% CI include 1.0, see *Table 2*, *Figure 3—figure supplement 2*).

Due to a lack of overlap in variables between datasets, we were unable to externally validate the 10-variable version of our growth faltering CPR. However, the top 2 predictors were available in both the GEMS and MAL-ED datasets. Therefore, we took the 2-variable CPR of growth faltering derived from children 0–23 months of age in GEMS, including HAZ at enrollment and age (the top 2 predictors), and externally validated it in MAL-ED data. The CPR had marginal discrimination in the GEMS data (AUC = 0.64, 95% CI: 0.64, 0.64), and a slight increase in discriminative ability at external validation in MAL-ED data (AUC = 0.68, 95% CI: 0.63, 0.74). On average, the CPR overestimated probability of growth faltering (calibration intercept −0.32, 95% CI: −0.54, −0.11), and predictions tended to be too moderate (calibration slope 1.5, 95% CI: 1.0, 2.1) (*Table 2*, *Figure 3*, *Figure 3—figure supplement 3*). Odds ratios for the 10-variable model predicting severe growth faltering in MAL-ED are shown in *Supplementary file 1*.

## Addition of MUAC, diarrhea etiology, and antibiotic use did not meaningfully impact discriminative performance of CPR to identify children who went on to severe growth faltering following acute diarrhea in GEMS

*Table 1* and *Supplementary file 1* present the results of the growth faltering sensitivity analyses. Top predictor variables were highly consistent across models and included patient demographics, patient symptoms, and indicators of household wealth. CPRs of higher age strata had higher AUCs (0.76 [95% CI: 0.75, 0.77] in 24–59 months in GEMS vs. 0.60 [95% CI: 0.59, 0.60] in 0–11 months in GEMS).

When MUAC was considered as a potential predictor (instead of HAZ), MUAC replaced HAZ as a top predictor, all other top 10 predictors remained the same, and AUC decreased (down to 0.70, 95% CI: 0.70, 0.70). When both HAZ and MUAC were considered as potential predictors, both were top predictors, but AUC remained unchanged compared to the main model that considered only HAZ (0.72, 95% CI: 0.72, 0.73). The predictors of very severe growth faltering (≥1.0 decrease in HAZ) were similar to the predictors of severe growth faltering (≥0.5 decrease in HAZ), though predictive ability was better (AUC 0.80 [95% CI: 0.79, 0.80] for ≥1.0 vs. 0.72 [95% CI: 0.71, 0.73] for ≥0.5).

Accounting for seasonality did not meaningfully improve the CPR, and antibiotic use and diarrhea etiology were consistently not ranked as top predictors of growth faltering (*Supplementary file 1*). The addition of more predictor variables only marginally improved AUCs. When data were limited to only those children not already stunted at initially presentation, top predictors and AUC of growth faltering were similar (0.71, 95% CI: 0.70, 0.72). While the top predictor variables were similar, the CPR predicting any stunting at follow-up was noticeably higher, AUC = 0.90 (95% CI: 0.90, 0.91).

## Derivation of a CPR to identify children without diarrhea (controls) who went on to severe growth faltering using GEMS data

Top predictors of growth faltering were similar in non-diarrhea controls compared to cases in GEMS (*Table 1*), but predictive ability was higher (AUC 0.79 [95% CI: 0.78, 0.79] in controls vs. 0.72 [95% CI: 0.72, 0.72] in cases). Again, top predictors were consistent with previous models and included age,

HAZ at enrollment, respiratory rate, temperature, number of people living in household, breastfed, number of rooms used for sleeping, number of children under 60 months old who live in household, education level of primary caregiver, and number of other households that share same fecal waste disposal facility (e.g., latrine). The maximum AUC attained with the model was 0.79 (95% CI: 0.79, 0.80) with a model of 15 variables, while an AUC of 0.79 (95% CI: 0.78, 0.79) and 0.79 (95% CI: 0.79, 0.79) could be obtained with a CPR of 5 and 10 variables, respectively (*Figure 3—figure supplement 4*).

## Discussion

By utilizing data from two large multicenter clinical studies of pediatric diarrhea, we used a combination of machine learning and conventional regression methods to derive and validate CPRs for linear growth faltering. The discriminative performance of our CPR for growth faltering was remarkably similar between the two datasets (AUC = 0.72, 95% CI: 0.72, 0.72, based on GEMS 0–59 months; 0.68, 95% CI: 0.67, 0.69 based on MAL-ED 0–24 months). We were then able to externally validate a 2-variable version, which also had similar discriminative ability between the datasets (AUC 0.64–0.68 for 0–23 and 0–24 months in GEMS and MAL-ED, respectively). Our findings suggest the potential for a parsimonious prediction rule-guided algorithm to identify young children with acute diarrhea for appropriate triage and follow-up.

The limited number of studies that aim to identify children most likely to growth falter after acute diarrhea have resulted in CPRs with varying discriminative and generalizability. Our full CPRs were better at identifying growth faltering than (*Brander et al., 2019*) (AUC = 0.67, 95% CI: 0.64, 0.69), which was not externally validated, and worse than (*Hanieh et al., 2019*) (AUC = 0.85, 95% CI: 0.80, 0.90), which only used data from a single country. The top predictors of growth faltering identified by random forests in our analysis were consistent with existing knowledge of the drivers of growth faltering – child demographics, child symptoms, and indicators of household wealth. The top 2 variables (used in our parsimonious externally validated CPR) were age and baseline HAZ. However, despite the inclusion of markers of disease severity (temperature, respiratory rate, number of days of diarrhea), overall ability to predict growth faltering was moderate, and consideration of additional factors related to nature of disease (etiology, antibiotics) did not improve discriminative ability. This is consistent with previous analysis in GEMS data that found treating diarrhea with antibiotics generally did not prevent growth faltering (except for *Shigella* infections, *Nasrin et al., 2021*).

Furthermore, the similar incidence of growth faltering in diarrhea cases and matched controls (particularly in the youngest children), as well as the almost identical predictive variables and similar AUCs, suggests that the impact of a single episode of acute diarrhea on growth trajectory may be relatively low. It is possible that the entire diarrheal history of a child (e.g., frequency and severity of acute diarrhea), or subclinical enteric infections that do not result in diarrhea, are more important to their growth trajectory than a single diarrheal episode, though evidence is mixed (*Checkley et al., 2008*; *Rogawski et al., 2018*; *Deichsel et al., 2020*). Indeed, while the design of GEMS does not allow for the exploration of this hypothesis, MAL-ED does. Total days in all diarrheal episodes, days with diarrhea so far this episode, and days since last diarrhea episode were all top 10 predictors of growth faltering in MAL-ED. Furthermore in GEMS, the average baseline HAZ at enrollment was 0.5 HAZ lower in children who did not experience growth faltering than in children who did (*Figure 3—figure supplement 5*), suggesting the possibility that children need to have high enough HAZ in order to have the potential to falter. In contrast, children enrolled in Mali had the highest median HAZ at enrollment, and also had the lowest proportion of children who experienced growth faltering (*Supplementary file 1*). It is also possible that the underlying cause(s) of stunting are complex and interrelated, and relatively simple predictive models are not able to accurately parse apart which children do and do not experience sufficient causes. In sensitivity analyses, we demonstrated our ability to predict any stunting at follow-up with high accuracy (*Table 1*, *Supplementary file 1*). However, this represents a related but distinct outcome from our original aim, namely a *slowing down* of growth as opposed to stunting, and may warrant different clinical intervention.

While effective interventions exist for treating acute malnutrition (e.g., exclusive breastfeeding for the first 6 months of life, inpatient- and community-based management of acute malnutrition using corn-soy blend or ready-to-use therapeutic food; *WHO, 2013*; *Bergeron and Castleman, 2012*; *Keats et al., 2021*), there are few evidence-based guidance on how to reverse the effects of chronic

malnutrition once a child is stunted (*Bergeron and Castleman, 2012*; *Leroy et al., 2015*; *Reinhardt and Fanzo, 2014*; *Pavlinac et al., 2018*; *WHO, 2015*). We found that approximately one in five children experience severe growth faltering subsequent to acute diarrhea, that is, an *additional* ≥0.5 decrease in HAZ in the 2–3 months after acute diarrhea. Currently, presenting to care for an acute illness, such as diarrhea, offers an opportunity for medical personnel to assess and treat children for acute malnutrition through intensive feeding programs. Our CPR provides a tool for identifying patients likely to experience additional growth faltering after acute diarrhea. Current malnutrition recommendations are based on patient presentation – whether a child is underweight when they present to the clinic. Our CPR could be used to identify children not currently stunted and therefore not currently recommended for nutritional interventions, but who are likely to slow down in growth and therefore at higher risk of incident stunting. Identifying these children would allow clinicians to connect patients with community-based nutrition interventions (e.g., maternal support for safe introduction of weening foods, small quantity lipid nutrient supplements, etc.; *Bhutta et al., 2013*; *Bhutta et al., 2008*; *Cole, 2020*; *Zhang et al., 2021*) to prevent *additional* effects of chronic malnutrition, namely irreversible stunting. Given our ability to predict growth faltering in healthy controls in GEMS, community screening for those at risk of growth faltering (not just those presenting with acute diarrhea) may also be prudent. This would represent a different potential intervention strategy and future research should explore this possibility further.

Our study has a number of strengths and limitations. We derived CPRs for growth faltering from two multisite, prospective studies that included longitudinal follow-up with extensive etiologic testing. Unlike previous work in this area, we used random forests for variable selection which do not require assumptions about relationships between the underlying variables and generally outperform (*Singal et al., 2013*) conventional model building techniques. While our complete-case analysis strategy could introduce bias due to missing data, we were able to re-derive the 10-variable version in two distinct datasets with similar results. While we were only able to externally validate a 2-variable version of our growth faltering CPR, its discriminative performance was similar to the full 10-variable version, and was robust to external validation. Furthermore, while the observation windows were large for many variables in the MAL-ED dataset used for external validation (up to 90 days for dietary variables, and up to 6 months for household descriptors), the variables of interest in the 2-variable CPR were observed no more than 31 days from the start of diarrhea. In addition, we considered all diarrhea as an outcome of interest in MAL-ED, whereas the analysis in GEMS was limited to MSD. When limiting the MAL-ED analysis to MSD as defined in GEMS, the top predictors and discriminative ability were very similar. The quasi-external validation between continents within GEMS data, as well as the country-specific models within GEMS, all had similar top predictors and discriminative performance, further supporting the overall validity of our CPR. Finally, we explored a range of AFe cutoffs for etiology, with consistent results.

Our study can also serve as a guide for future CPR development. We used a prediction-based approach for variable selection, and compared multiple model fitting strategies. We assessed model calibration as well as discrimination, and reported the results from numerous sensitivity analyses. Finally, we designed our study *a priori* to incorporate external validation, lending additional confidence to the generalizability of our results.

In conclusion, we used data from two large multi-country studies to derive and validate a CPR for growth faltering in children presenting for diarrhea treatment. Our findings indicate that use of prediction rules, potentially applied as clinical decision support tools, could help to identify additional children at risk of poor outcomes after an episode of diarrheal illness, that is not currently stunted but likely to decelerate growth. In settings with high mortality and morbidity in early childhood, such tools could represent a cost-effective way to target resources toward those who need it most.

## Data availability

GEMS and MAL-ED data are available to the public by request through the following website https://clinepidb.org/ce/app/. Data cleaning and statistical code needed to reproduce all parts of this analysis are available from the corresponding author's GitHub page: https://github.com/LeungLab/CPRgrowthfaltering, (copy archived at swh:1:rev:f3fd53b5713ef787d3ae2cd4a81f3286f52f2746, *Ahmed, 2022*).

## Acknowledgements

This work was supported by National Institutes of Health under Ruth L Kirschstein National Research Service Award NIH T32AI055434 and by the National Institute of Allergy and Infectious Diseases (R01AI135114). The funders had no role in study design, data collection, and interpretation, or the decision to submit the work for publication.

## Additional information

### Funding

| Funder | Grant reference number | Author |
| --- | --- | --- |
| National Institute of Allergy and Infectious Diseases | R01AI135114 | Sharia M Ahmed<br>Ben J Brintz<br>Daniel T Leung |
| National Institutes of Health | T32AI055434 | Sharia M Ahmed |

The funders had no role in study design, data collection, and interpretation, or the decision to submit the work for publication.

### Author contributions

Sharia M Ahmed, Conceptualization, Resources, Data curation, Software, Formal analysis, Validation, Investigation, Methodology, Writing - original draft, Writing - review and editing; Ben J Brintz, Software, Formal analysis, Validation, Methodology, Writing - review and editing; Patricia B Pavlinac, Adam C Levine, Eric J Nelson, James A Platts-Mills, Karen L Kotloff, Conceptualization, Resources, Writing - review and editing; Lubaba Shahrin, Sayeeda Huq, Conceptualization, Writing - review and editing; Daniel T Leung, Conceptualization, Resources, Supervision, Funding acquisition, Writing - review and editing

### Author ORCIDs

Sharia M Ahmed ⓘ http://orcid.org/0000-0002-4060-5712
Ben J Brintz ⓘ http://orcid.org/0000-0003-4695-0290
Daniel T Leung ⓘ http://orcid.org/0000-0001-8401-0801

### Ethics

Parents or caregivers of participants provided informed consent, either in writing or witnessed if parents or caregivers were illiterate. The GEMS study protocol was approved by ethical review boards at each field site and the University of Maryland, Baltimore, USA. The MAL-ED study protocol was approved by ethical review boards at each field site and the Johns Hopkins Institutional Review Board, Baltimore, USA. This analysis utilized publicly available data from both studies, see Data Availability statement, and as such is non-human subjects research.

### Decision letter and Author response

Decision letter https://doi.org/10.7554/eLife.78491.sa1
Author response https://doi.org/10.7554/eLife.78491.sa2

## Additional files

### Supplementary files
• MDAR checklist
• Supplementary file 1. Additional methodological details and results of sensitivity analyses.

### Data availability

This is a secondary analysis of the GEMS and MAL-ED datasets. These data are available to the public through the following website https://clinepidb.org/ce/app/. Data requests are submitted through the website listed, and requests are reviewed and approved by the investigators of those original

studies consistent with their protocols and data sharing policies. As of the time of submission of this manuscript, the GEMS Data Access Request asked for purpose, hypothesis/research question, analysis plan, dissemination plan, and if anyone from the GEMS study team had already been approached regarding this request. The MAL-ED data were available for download without submitting a Data Access Request. Data cleaning and statistical code needed to reproduce all parts of this analysis are available from the corresponding author's GitHub page: https://github.com/LeungLab/CPRgrowthfaltering, (copy archived at swh:1:rev:f3fd53b5713ef787d3ae2cd4a81f3286f52f2746). The following previously published datasets were used: Dataset 1: Gates Enterics Project, Levine MM, Kotloff K, Nataro J, Khan AZA, Saha D, Adegbola FR, Sow S, Alonso P, Breiman R, Sur D, Faruque A. 2018. Study GEMS1 Case Control. https://clinepidb.org/ce/app/record/dataset/DS_841a9f5259#Contacts. Database and Identifier: ClinEpiDB, DS_841a9f5259 Dataset 2: The Etiology, Risk Factors, and Interactions of Enteric Infections and Malnutrition and the Consequences for Child Health (MAL-ED). Primary Contact: David Spiro, Fogarty International Center, National Institutes of Health, Bethesda, MD, USA. https://clinepidb.org/ce/app/workspace/analyses/DS_5c41b87221/new/details.

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
