## [Editor Report]

This work would be valuable to global health scientists, particularly in low- and middle-income countries where childhood stunting is an ongoing challenge, and to statisticians interested in building clinical prediction rules. The authors' solid methodology leveraged large, rich datasets from multi-center studies to build and validate predictive models.

---

## [Decision Letter]

**Decision letter after peer review:**

Thank you for submitting your article "Derivation and external validation of clinical prediction rules identifying children at risk of linear growth faltering (stunting) presenting for diarrheal care" for consideration by *eLife*. Your article has been reviewed by 2 peer reviewers, and the evaluation has been overseen by me in my joint role as Reviewing Editor and Senior Editor. The following individual involved in review of your submission has agreed to reveal their identity: Andrew N Mertens (Reviewer #2).

As is customary in *eLife*, the reviewers have discussed their critiques with one another and with the Editors, and I prepared this decision letter to help you prepare a revised submission. Given the extent of the suggestions, I prefer to provide you with a compilation of the relevant suggestions in the two critiques to assist you in preparing the revisions for an eventual resubmission.

Essential revisions:

Although we expect that you will address these comments in your response letter, we also need to see the corresponding revision clearly marked in the text of the manuscript. Some of the reviewers' comments may seem to be simple queries or challenges that do not prompt revisions to the text. Please keep in mind, however, that readers may have the same perspective as the reviewers. Therefore, it is essential that you amend or expand the text to clarify the narrative accordingly.

The outcome used for prediction in a binary indicatory for a decrease in height-for-age Z-score >= 0.5. A child who fails to gain height by future measurements is of concern, but this outcome also misses children who are already experiencing growth failure, and is vulnerable to regression to the mean effect. The two most important predictors were age and current size, with current size having a positive association with risk of growth faltering. As mentioned in the discussion, there is "the possibility that children need to have high enough HAZ in order to have the potential to falter." Additionally, there may be children with erroneously high height measurements at the first measurement, so that the HAZ change >= 0.5 associated with high baseline HAZ is from measurement-error regression to the mean. I recommend also predicting absolute HAZ (or stunting status) as a secondary outcome and comparing if the important predictors change. Were alternative specifications (e.g., quantitative decrease in HAZ, incident stunting) considered?

In its current form, the results and conclusions from the results have problematic implications for the treatment of child malnutrition. The conclusion states: "In settings with high mortality and morbidity in early childhood, such tools could represent a cost-effective way to target resources towards those who need it most." If the current CPR was used in a resource-constrained setting, it would recommend that larger children should be prioritized for nutritional supplementation over already stunted children who may have reached their growth faltering floor. In addition, with a sensitivity of 80%, the tool would miss treating a large number of children who would experience growth faltering. The results of the clinical prediction tool need to be presented with care in how it could be used to prioritize treatment without missing treating children who would benefit from nutritional supplementation. Including absolute HAZ as an outcome will help, along with additional discussion of how the CPR fits alongside current treatment recommendations. For example, does this rule indicate treating children who aren't currently treated, or are there children who don't need treatment given current guidelines and the created CPR.

The results from these datasets may not have identified novel and strong predictors of growth faltering, as the current results indicate that additional predictors beyond current size and age don't help with predictions, but the analysis could be reframed as a template for developing a CPR, using this data as a case study. If age and current size are the only important predictors, then a simple rule based on age and a current HAZ cutoff could be created, negating the need for a more complicated model, but this manuscript also provides a good template for other clinical prediction analyses. Could you comment more on the methods and performance metrics used in the discussion, and make a recommendation for future analyses?

Why use the MAL-ED data to externally validate the CPR developed using GEMS data, given the different study designs, definitions of diarrheal disease, and predictors measured. Because GEMS is a multisite study, wouldn't it be easier, and allow more complex models to be validated, if the model was fit using data from some countries, then validated in populations from other countries?

In addition to the coefficients for the 10-variable model, it would be helpful to present coefficients for the final 2-variable model that was assessed in both GEMS and MAL-ED.

Although the authors opted to use logistic regression based on AUC, the AUC values for random forest models were only slightly lower (Figure S2), and random forest may provide simpler clinical prediction rules. It may be interesting to also describe the rules that were developed by the random forest models. The last panel in Figure S2 may be mislabeled (0-23 mo for MAL-ED instead of 0-59 mo).

I am not very familiar with the variable importance calculated from random forest models. What is the implication of certain features having high variable importance, but also having coefficient estimates that are indistinguishable from the null (e.g., age in MAL-ED, respiratory rate in GEMS in Table S4)?

In the Discussion (p.20), the authors note that the entire diarrheal history of a child may be a more important indicator of linear growth faltering than a single episode. These datasets seem potentially well-suited to directly explore this question ¬- were frequency/number of prior diarrheal episodes investigated as predictors in GEMS / MAL-ED?

For reproducibility, please specify the software and key packages with corresponding versions that were used for this analysis.

The best performing model was logistic regressions fit with variables chosen by random forest models. Any idea why this would be? Is it because they are simpler and the random forest models are overfit to the training data? I would expect them to perform worse because they don't allow for nonlinearity and interactions like a RF model. If generalized linear models perform better than random forest for prediction in this situation, penalized logistic regression models may also improve predictive performance by incorporating variable selection with prediction in a simpler model than random forests.

The conclusion in the abstract is "Our findings indicate that use of prediction rules could help identify children at risk of poor outcomes after an episode of diarrheal illness", but prediction performance is the same in control children, so while its important to retain the discussion of lack of association between diarrhea and growth, the framing of the paper could be expanded around all children in LMIC, rather than just children with acute diarrhea. This could just be a slight reframing in the writing, or you could expand the MAL-ED prediction model to use all children in addition to the prediction on the subset of children with diarrhea.

What is the rationale for comparing HAZ and MUAC as separate and combined predictors of growth? On one hand, it's interesting to compare which current measures of anthropometry are most associated with future measures of anthropometry, in which case you'd want to include other outcomes such as WHZ, WAZ, and MUAC. But if the goal is to develop the best clinical prediction tool, it makes more sense to include all measures of growth that can be easily clinically collected as predictors to see if performance increases by including WHZ, WAZ, and MUAC on top of HAZ.

Line 125-128: "Model performance was assessed using the receiver operating characteristic (ROC) curves and the cross-validated C-statistic (area under the ROC curve (AUC)), a measures which describes how well a model can discriminate between the two outcomes, from the cross-validation." Confusingly worded… do you mean "AUC is a measure which describes how well a model can predict a binary outcome in test data from the cross-validated folds."

Line 129-142: Model calibration performance metrics: these were new to me, and I wasn't sure what to be looking for or what story they could tell us about model performance beyond the AUC. What is the reader looking for? Can they tell us something different than the AUC?

Line 173: separately report missing versus implausible values, because the percent implausible gives an indication of data quality.

Lines 177-182: Report mean HAZ by country as well to show if it there is lower growth faltering in some countries because of high existing stunting by the age of first measurement.

Line 199: This is the first mention of death as an outcome (and the results of the CPR for death are not discussed).

Page 20: "It is possible that the entire diarrheal history of a child (e.g. frequency and severity of acute diarrhea), or subclinical enteric infections that do not result in diarrhea, are more important to their growth trajectory than a single diarrheal episode, though evidence is mixed." As you have longitudinal data from MAL-ED, can't you explicitly check this by using diarrhea history as a predictor?

Page 21: "Unlike previous work in this area, we used random forests for variable selection which do not require assumptions about the underlying variables and generally outperform(49) conventional model building techniques."

– Need to clarify that random forests have no assumptions about the relationship between variables, not about the variables themselves, which still have assumptions around how they are coded/categorized.

Tables S4- Age is the most important predictor, but the OR is 1 with 1,1 confidence intervals. Can you convert the predictor to age in months or report more decimal places so direction of effect can be seen?

---

## [Author Response]

Essential revisions:Although we expect that you will address these comments in your response letter, we also need to see the corresponding revision clearly marked in the text of the manuscript. Some of the reviewers' comments may seem to be simple queries or challenges that do not prompt revisions to the text. Please keep in mind, however, that readers may have the same perspective as the reviewers. Therefore, it is essential that you amend or expand the text to clarify the narrative accordingly.The outcome used for prediction in a binary indicatory for a decrease in height-for-age Z-score >= 0.5. A child who fails to gain height by future measurements is of concern, but this outcome also misses children who are already experiencing growth failure, and is vulnerable to regression to the mean effect. The two most important predictors were age and current size, with current size having a positive association with risk of growth faltering. As mentioned in the discussion, there is "the possibility that children need to have high enough HAZ in order to have the potential to falter." Additionally, there may be children with erroneously high height measurements at the first measurement, so that the HAZ change >= 0.5 associated with high baseline HAZ is from measurement-error regression to the mean. I recommend also predicting absolute HAZ (or stunting status) as a secondary outcome and comparing if the important predictors change. Were alternative specifications (e.g., quantitative decrease in HAZ, incident stunting) considered?

Thank you for this suggestion. We have added additional models for the following predictions: (a) growth faltering in those NOT stunted (HAZ≥-2) at presentation, (b) any stunting (HAZ<-2) at follow-up, and (c) any stunting at follow-up in those not stunted at presentation.

While we agree the addition of these models improves the manuscript, we also want to highlight that these models have distinct outcomes and therefore have separate clinical uses. Our original goal was to identify children whose growth was likely to slow down after diarrhea. As we show, top predictors and predictive performance is similar for growth faltering across baseline stunting status. We present any stunting at follow-up as a comparison, but argue that this is a different clinical outcome that may warrant different intervention.

P.22 L.335-339: “In sensitivity analyses, we demonstrated our ability to predict any stunting at follow-up with high accuracy (Table 1, Supplementary file 1E). However, this represents a related but distinct outcome from our original aim, namely a slowing down of growth as opposed to stunting, and may warrant different clinical intervention.”

P.23 L.349-353: “Current malnutrition recommendations are based on patient presentation – whether a child is underweight when they present to the clinic. Our CPR could be used to identify children not currently stunted and therefore not currently recommended for nutritional interventions, but who are likely to slow down in growth and therefore at higher risk of incident stunting.”

In its current form, the results and conclusions from the results have problematic implications for the treatment of child malnutrition. The conclusion states: "In settings with high mortality and morbidity in early childhood, such tools could represent a cost-effective way to target resources towards those who need it most." If the current CPR was used in a resource-constrained setting, it would recommend that larger children should be prioritized for nutritional supplementation over already stunted children who may have reached their growth faltering floor. In addition, with a sensitivity of 80%, the tool would miss treating a large number of children who would experience growth faltering. The results of the clinical prediction tool need to be presented with care in how it could be used to prioritize treatment without missing treating children who would benefit from nutritional supplementation. Including absolute HAZ as an outcome will help, along with additional discussion of how the CPR fits alongside current treatment recommendations. For example, does this rule indicate treating children who aren't currently treated, or are there children who don't need treatment given current guidelines and the created CPR.

We thank the Reviewers for pointing out this oversight. We have edited the Discussion for clarity as follows.

P.23 L.348-357: “Our CPR provides a tool for identifying patients likely to experience additional growth faltering after acute diarrhea. Current malnutrition recommendations are based on patient presentation – is a child underweight when they come to the clinic. Our CPR could be used to identify children not currently stunted and therefore not currently recommended for nutritional interventions, but who are likely to slow down in growth and therefore at higher risk of incident stunting. Identifying these children would allow clinicians to connect patients with community-based nutrition interventions (e.g. maternal support for safe introduction of weening foods, small quantity lipid nutrient supplements (SQ-LNS), etc. (45-48)) to prevent additional effects of chronic malnutrition, namely irreversible stunting.”

P.25 L.386-389: “Our findings indicate that use of prediction rules, potentially applied as clinical decision support tools, could help to identify additional children at risk of poor outcomes after an episode of diarrheal illness, i.e. not currently stunted but likely to decelerate growth.”

The results from these datasets may not have identified novel and strong predictors of growth faltering, as the current results indicate that additional predictors beyond current size and age don't help with predictions, but the analysis could be reframed as a template for developing a CPR, using this data as a case study. If age and current size are the only important predictors, then a simple rule based on age and a current HAZ cutoff could be created, negating the need for a more complicated model, but this manuscript also provides a good template for other clinical prediction analyses. Could you comment more on the methods and performance metrics used in the discussion, and make a recommendation for future analyses?

Thank you for this suggested. We have edited the Discussion as below:

P.24 L.380-384: “Our study can also serve as a guide for future CPR development. We used a prediction-based approach for variable selection, and compared multiple model fitting strategies. We assessed model calibration as well as discrimination, and reported the results from numerous sensitivity analyses. Finally, we designed our study a priori to incorporate external validation, lending additional confidence to the generalizability of our results.”

Why use the MAL-ED data to externally validate the CPR developed using GEMS data, given the different study designs, definitions of diarrheal disease, and predictors measured. Because GEMS is a multisite study, wouldn't it be easier, and allow more complex models to be validated, if the model was fit using data from some countries, then validated in populations from other countries?

We thank the Reviewers for this suggestion and have added country-specific CPRs in the Supplement. We have also added a sensitivity analysis whereby we fit models to all data from one continent in GEMS, and then validated that model on the other continent in GEMS data. As you can see from Supplementary file 1E top predictors and discriminative performance were similar across countries and continents

P.10 L.171-173: “Finally, we conducted a quasi-external validation within the GEMS data by fitting a model to one continent and validating it on the other.”

P.24 L.376-379: “The quasi-external validation between continents within GEMS data, as well as the country-specific models within GEMS, all had similar top predictors and discriminative performance, further supporting the overall validity of our CPR. Finally, we explored a range of AFe cutoffs for etiology, with consistent results.”

In addition to the coefficients for the 10-variable model, it would be helpful to present coefficients for the final 2-variable model that was assessed in both GEMS and MAL-ED.

We have added this as requested to the Supplementary file 1D.

Although the authors opted to use logistic regression based on AUC, the AUC values for random forest models were only slightly lower (Figure S2), and random forest may provide simpler clinical prediction rules. It may be interesting to also describe the rules that were developed by the random forest models. The last panel in Figure S2 may be mislabeled (0-23 mo for MAL-ED instead of 0-59 mo).

Thank you for the correction, we have edited the figure as appropriate. All of our Results are presented in our manuscript and Supplement, we did not develop additional CPRs.

I am not very familiar with the variable importance calculated from random forest models. What is the implication of certain features having high variable importance, but also having coefficient estimates that are indistinguishable from the null (e.g., age in MAL-ED, respiratory rate in GEMS in Table S4)?

We appreciate this is a very important question. As described on P.7 L116-118, we defined variable importance as mean squared prediction error achieved by including the variable in the predictive model. In other words, we selected variables based on how well they improved the predictive performance of the overall model. This is a different analytic goal than testing the hypothesis that a variable is or is not associated with the outcome (e.g. does the confidence interval for the odds ratio cross 1). If our goal had been to explore the association between potential risk factors and growth faltering, we would have implemented a different variable selection process. Please refer to Shmueli 2010 and van Diepen 2017 for additional details.

In the Discussion (p.20), the authors note that the entire diarrheal history of a child may be a more important indicator of linear growth faltering than a single episode. These datasets seem potentially well-suited to directly explore this question ¬- were frequency/number of prior diarrheal episodes investigated as predictors in GEMS / MAL-ED?

We thank the Reviewers for bringing this omission to our attention. The study design of GEMS does not make it possible to assess history of previous diarrhea episodes. There are a number of variables that approximate this in MAL-ED, which have already been considered as potential predictors in the re-derivation. We have the Discussion as follows to incorporate this.

P.22 L.322-328: “It is possible that the entire diarrheal history of a child (e.g. frequency and severity of acute diarrhea), or subclinical enteric infections that do not result in diarrhea, are more important to their growth trajectory than a single diarrheal episode, though evidence is mixed (13, 26, 37). Indeed, while the design of GEMS does not allow for the exploration of this hypothesis, MAL-ED does. Total days in all diarrheal episodes, days with diarrhea so far this episode, and days since last diarrhea episode were all top-10 predictors of growth faltering in MAL-ED.”

For reproducibility, please specify the software and key packages with corresponding versions that were used for this analysis.

Thanks for this suggestion, we have added as below.

P.10 L.173-174: “All analysis was conducted in R 4.0.2 using the packages “ranger,” “cvAUC,” and “pROC.””

The best performing model was logistic regressions fit with variables chosen by random forest models. Any idea why this would be? Is it because they are simpler and the random forest models are overfit to the training data? I would expect them to perform worse because they don't allow for nonlinearity and interactions like a RF model. If generalized linear models perform better than random forest for prediction in this situation, penalized logistic regression models may also improve predictive performance by incorporating variable selection with prediction in a simpler model than random forests.

We agree that exploring alternative model building strategies could prove fruitful. However, incorporating variable selection and prediction into a single model building process such as with ridge regression or elastic net could lead to a more complicated final model, as less important coefficients approach but do not reach 0. In any case, an exhaustive comparison between model building strategies was beyond the scope of this study.

The conclusion in the abstract is "Our findings indicate that use of prediction rules could help identify children at risk of poor outcomes after an episode of diarrheal illness", but prediction performance is the same in control children, so while its important to retain the discussion of lack of association between diarrhea and growth, the framing of the paper could be expanded around all children in LMIC, rather than just children with acute diarrhea. This could just be a slight reframing in the writing, or you could expand the MAL-ED prediction model to use all children in addition to the prediction on the subset of children with diarrhea.

We thank the Reviewer for this suggestion. We feel it is important to retain the focus on acute diarrhea, as this represents an easy point of access to identify children who are struggling. A community screening program that would utilize a CPR predicting growth faltering in both symptomatic and non symptomatic children could also be beneficial, but is a different goal and would fit in a different type of intervention than our original research goal. Per your suggestion, we have edited the Abstract and Discussion as follows to highlight this possibility.

P.2: “Abstract Conclusions: Our findings indicate that use of prediction rules could help identify children at risk of poor outcomes after an episode of diarrheal illness. They may also be generalizable to all children, regardless of diarrhea status.”

P.23 L357-360: “Given our ability to predict growth faltering in healthy controls in GEMS, community screening for those at risk of growth faltering (not just those presenting with acute diarrhea) may also be prudent. This would represent a different potential intervention strategy and future research should explore this possibility further.”

What is the rationale for comparing HAZ and MUAC as separate and combined predictors of growth? On one hand, it's interesting to compare which current measures of anthropometry are most associated with future measures of anthropometry, in which case you'd want to include other outcomes such as WHZ, WAZ, and MUAC. But if the goal is to develop the best clinical prediction tool, it makes more sense to include all measures of growth that can be easily clinically collected as predictors to see if performance increases by including WHZ, WAZ, and MUAC on top of HAZ.

Our goal was to develop the best clinical prediction tool in terms of predictive ability AND ease of use. As we show in Supplementary file 1E, the model considering HAZ as the only growth metric performed better than the model considering only MUAC, and performed just as well as the model considering both. Therefore, we concluded the most predictive and parsimonious model included HAZ as the only growth metric. Because our goal was to develop a clinical prediction rule for pediatric patients with acute diarrhea, we chose not to consider WHZ and WAZ. Child weight is highly susceptible to dehydration status, especially in the youngest children (Modi 2015). Therefore, weight-based growth metrics can be highly inaccurate during acute diarrheal illness.

Line 125-128: "Model performance was assessed using the receiver operating characteristic (ROC) curves and the cross-validated C-statistic (area under the ROC curve (AUC)), a measures which describes how well a model can discriminate between the two outcomes, from the cross-validation." Confusingly worded… do you mean "AUC is a measure which describes how well a model can predict a binary outcome in test data from the cross-validated folds."

Thanks for this suggestion, edits below

P.8 L.125-128: “Model performance was assessed using the receiver operating characteristic (ROC) curves and the cross-validated C-statistic (area under the ROC curve (AUC)). The AUC describes how well a model can discriminate between a binary outcome in the test data from the cross-validated folds.”

Line 129-142: Model calibration performance metrics: these were new to me, and I wasn't sure what to be looking for or what story they could tell us about model performance beyond the AUC. What is the reader looking for? Can they tell us something different than the AUC?

Model discrimination (e.g. AUC) and model calibration are indeed two different metrics for evaluating predictive performance. Discrimination refers to a model’s ability to correctly separate who does and does not experience the outcome of interest. Calibration refers to a model’s ability to correctly estimate the risk of the outcome. In other words, how similar is the predicted number of events to the observed number of events. We have added the following brief explanation to the manuscript and included an addition citation for interested readers.

P.8 L.129-130: “Calibration refers to a model’s ability to correctly estimate the risk of the outcome(34). We assessed model calibration both quantitatively and graphically… “

Line 173: separately report missing versus implausible values, because the percent implausible gives an indication of data quality.

We have edited as follows.

P.10 L.177-180: “There were 9439 children with acute diarrhea enrolled in GEMS. In the analysis of the primary outcome (growth faltering), 110 observations were dropped for having follow-up measurements taken <49 or >91 days after enrollment, and 1276 were dropped for having implausible HAZ measurements, leaving an analytic sample of 8053.”

Lines 177-182: Report mean HAZ by country as well to show if it there is lower growth faltering in some countries because of high existing stunting by the age of first measurement.

We thank the Reviewers for this excellent suggestion, and have added the requested data in Supplementary file 1B. We have edited the Discussion as follows.

P.22 L.328-333: “Furthermore, the average baseline HAZ at enrollment was 0.5 HAZ lower in children who did not experience growth faltering than in children who did (Supplemental Figure S4), suggesting the possibility that children need to have high enough HAZ in order to have the potential to falter. In contrast, children enrolled in Mali had the highest median HAZ at enrollment, and also had the lowest proportion of children who experienced growth faltering (Supplementary file 1B).”

Line 199: This is the first mention of death as an outcome (and the results of the CPR for death are not discussed).

This was an oversight from a previous draft of the manuscript and has been removed. Thanks for pointing this out.

Page 20: "It is possible that the entire diarrheal history of a child (e.g. frequency and severity of acute diarrhea), or subclinical enteric infections that do not result in diarrhea, are more important to their growth trajectory than a single diarrheal episode, though evidence is mixed." As you have longitudinal data from MAL-ED, can't you explicitly check this by using diarrhea history as a predictor?

Please see response and edits listed above.

Page 21: "Unlike previous work in this area, we used random forests for variable selection which do not require assumptions about the underlying variables and generally outperform(49) conventional model building techniques."– Need to clarify that random forests have no assumptions about the relationship between variables, not about the variables themselves, which still have assumptions around how they are coded/categorized.

Thank you for pointing this out, we have edited as follows.

P.24 L.363-365: “Unlike previous work in this area, we used random forests for variable selection which do not require assumptions about relationships between the underlying variables and generally outperform(50) conventional model building techniques.”

Tables S4- Age is the most important predictor, but the OR is 1 with 1,1 confidence intervals. Can you convert the predictor to age in months or report more decimal places so direction of effect can be seen?

As discussed earlier in this Response to Reviewers, the goal of our CPRs was prediction, not to assess the association or effect of a risk factor on the outcome. Therefore, it is inappropriate, i.e. the analytic strategy does not support, to judge the relationship between risk factors and the outcome using these CPR models.

References

Shmueli, G. To explain or to predict? Statist. Sci. 25(3): 289-310 (August 2010). DOI:10.1214/10-STS330. https://arxiv.org/pdf/1101.0891.pdf.

van Diepen, M., Ramspek, C.L., Jager, K.J., Zoccali, C., Dekker, Friedo W. Predictionversus aetiology: common pitfalls and how to avoid them. Nephrol DialTransplant. 2017 Apr 1;32. PMID: 28339854.

Modi, P., Nasrin,S., Hawes, M., Glavis-Bloom, J., Alam N.H., Hossain, M.I., Levine, A.C. Midupper arm circumference outperforms weight-based measures of nutritionalstatus in children with diarrhea. J. Nutr. 2015 Jul; 145(7):1582-7. PMID:25972523.